# Stakeholder theory and management: Understanding longitudinal collaboration networks

**Julian Fares**[1]*, **Kon Shing Kenneth Chung**[2], **Alireza Abbasi**[3]

**1** Department of Management Studies, Adnan Kassar School of Business, Lebanese American University, Beirut, Lebanon, **2** School of Project Management, Faculty of Engineering, The University of Sydney, Sydney, Australia, **3** School of Engineering and IT, University of New South Wales (UNSW), Canberra, Australia

\* Julian.fares@lau.edu.lb

**Data Availability Statement:** All relevant data are within the paper and its Supporting information files.

**Funding:** The author(s) received no specific funding for this work.

## Abstract

This paper explores the evolution of research collaboration networks in the 'stakeholder theory and management' (STM) discipline and identifies the longitudinal effect of co-authorship networks on research performance, i.e., research productivity and citation counts. Research articles totaling 6,127 records from 1989 to 2020 were harvested from the Web of Science Database and transformed into bibliometric data using Bibexcel, followed by applying social network analysis to compare and analyze scientific collaboration networks at the author, institution and country levels. This work maps the structure of these networks across three consecutive sub-periods ($t_1$: 1989–1999; $t_2$: 2000–2010; $t_3$: 2011–2020) and explores the association between authors' social network properties and their research performance. The results show that authors collaboration network was fragmented all through the periods, however, with an increase in the number and size of cliques. Similar results were observed in the institutional collaboration network but with less fragmentation between institutions reflected by the increase in network density as time passed. The international collaboration had evolved from an uncondensed, fragmented and highly centralized network, to a highly dense and less fragmented network in $t_3$. Moreover, a positive association was reported between authors' research performance and centrality and structural hole measures in $t_3$ as opposed to ego-density, constraint and tie strength in $t_1$. The findings can be used by policy makers to improve collaboration and develop research programs that can enhance several scientific fields. Central authors identified in the networks are better positioned to receive government funding, maximize research outputs and improve research community reputation. Viewed from a network's perspective, scientists can understand how collaborative relationships influence research performance and consider where to invest their decision and choices.

**Competing interests:** The authors have declared that no competing interests exist.

# Introduction

The emergence of research collaboration networks has largely contributed to the development of many scientific fields and the exponential increase in research publications [1]. Scientific collaboration is described as the interaction occurring between two or more entities (e.g. authors, institutions, countries) to advance a field of knowledge by uncovering scientific findings in more efficient ways that might not be possible through individual efforts [2, 3]. Collaborative relationships affect research performance by disseminating the flow of knowledge, improving research capacity, enhancing innovation, creating new knowledge sources, reducing research cost through economies of scope, and creating synergies between multi-disciplinary teams [2, 4–7]. Therefore, understanding the status quo of a scientific discipline requires understanding the social structure and composition of these collaborative relationships [1, 8, 9].

Social network analysis (SNA) is one of the most utilized methods for exploring scientific collaboration networks. SNA can quantify, analyze and visualize relationships in a specific research community, identify central opinion leaders that are leading collaborative works as well as evaluate the underlying structures that are influencing collaboration. Usually in a scientific collaboration network, the authors, institutions, and countries are referred to as "actors" or "nodes" and the collaborative relationships between them as "ties". Indeed, there are a plethora of studies that used SNA to examine scientific collaboration networks of co-authors in various disciplines [2, 10–18]. However, the findings of the above studies remain inconclusive regarding the longitudinal associations between structures of co-authorship networks and research performance across different sub-periods [18–20], and particularly in the "stakeholder theory and management" (STM) field, there is paucity of evidence. The value of the STM discipline in scientometrics and scientific collaboration research lies in its cross-disciplinary nature, i.e., having been applied in various business [21, 22] and non-business domains [23–25], interconnecting different scientific disciplines that were once considered dispersed. The stakeholder theory is considered by many as a "living Wiki"- that is continuously growing through the collaboration of various scholars from different research fields. In light of the above argument, the aims of this study are to:

1. explore the evolution of research collaboration networks of each of the authors, institutions, and countries in the STM discipline and across three consecutive sub-periods ($t_1$: 1989–1999; $t_2$: 2000–2010; $t_3$: 2011–2020),

2. identify the key actors (authors, institutions, and countries) that are leading collaborative works in each sub-period, and

3. understand the longitudinal effect of co-authorship networks on research performance measured by research productivity (i.e. the number of published papers) and citation counts of the entities [26].

Certainly, scholars can collaborate in a multitude of different ways ranging from faculty-based administrative works, conference participations, meetings, seminars, inter-institutional joint projects and informal relationships [27]. However, this study uses co-authorship analysis–as a widely used and reliable bibliometric method that explores co-authorship relationship on scientific papers between different actors (nodes) being authors, institutions or countries. Therefore, the analysis in this paper is carried out at three level: the micro level–authors of the same or different institutions; the meso level–inter-institutional strategic alliances (universities and departments); and the macro level–international partnerships entailing the authors and institutions, all of which are major spectrums of research collaboration [7, 28].

To do so, the web of science (WoS) database is used to extract the bibliometric data of 6127 journal articles published in the last 32 years (1989–2020). This data was analyzed using Bibexcel as a package program for bibliometric analysis, UCINET for further SNA, and VOSviewer for visualizing the networks. The results provide important insights for allocating governmental funding, maximizing research output, improving research community reputation and enhancing cost savings that all should be directly or indirectly piloted by the most suitable scientists that can influence and lead collaborative research in their networks [29, 30].

This paper starts with a brief history of STM research, followed by an overview of network theories most relevant to this study. Then, the methodology for data collection, refinement and analysis is described. Descriptive and SNA results are presented for each of the examined networks across the three sub-periods, followed by the findings of the association testing between different social network measures (ego-density, degree centrality, betweenness centrality, closeness centrality, efficiency, constraint and average tie strength) and each of the citation counts and research productivity metrics. Lastly, the conclusions and the theoretical and practical implications are provided.

## Literature review

### Origins of STM

The stakeholder concept was first originated in the Stanford Research Institute in the 1960s, and then more formally introduced by Freeman [31] as a new theory of strategic management that aims to create value for various organizational groups and individuals to achieve business success. The stakeholder theory aims to define and create value, interconnect capitalism with ethics and identify appropriate management practices [32]. A stakeholder is best defined as "any group or individual who can affect or is affected by the achievement of the organization's objectives" [31]. Freeman emphasized on the relationships between the organization and its stakeholders as the central unit of analysis and a point of departure for stakeholder research. Accordingly, Rowley [33] was the first to introduce social networks to STM to understand the mechanism of such relationships. In particular, he argued that a focal firm's response to stakeholder pressure is based on the interplay between the centrality of the focal firm and the density of stakeholder alliances. There have been many seminal works that put stakeholder theory on a solid managerial science footing, such that of Donaldson and Preston's [34] that conceptualized the theory from a descriptive, instrumental and normative approach, followed by Mitchell et al. [35] who proposed a framework for identifying stakeholder salience using the attributes of power, legitimacy and urgency, and so on [36–39].

### Expansion of STM

From the early 2000s, stakeholder theory has shown to be a class of management theory rather than an exclusive theory, per se, by its applicability in various business domains such as business ethics [40–42], finance [43–45], accounting [46, 47], marketing [22, 48, 49] and management [21, 50, 51]. Afterwards, the interest has moved to stakeholder analysis—a main systematical analytical process for stakeholder management that involves identifying and categorizing stakeholders, and identifying best practices for engaging them [52]. Even some scientific disciplines, such as project management, has considered stakeholder management as one of its core knowledge areas for achieving project success [53]. This exponential growth of the field has resulted in more than 55 stakeholder definitions [54] and numerous frameworks for stakeholder identification [35, 55, 56], categorization [57, 58], and engagement [59–62]. However, the enlargement of the stakeholder analysis body caused ambiguousness in its concepts and purpose [34, 56, 63], where it turned into an experimental field for different methods to be

explored. Jepsen and Eskerod [64] revealed that the tools used for stakeholder identification and categorization were not clear enough for project managers to use, being referred to as theoretical [65].

The theoretical debates seemed to have alleviated between 2010 and 2020, where researchers focused instead on the applicability of stakeholder theory in the real world cases [66, 67]. Empirical studies mainly examined the behavior of firms and their stakeholders towards each other, such as how firms manage stakeholders [68, 69] and how stakeholders influence a firm [70]. Once again, the scientific paradigm of STM has mostly been uncovered in the domains of strategic management [71, 72] and project management [73–75]. Therefore, it is evident that growth of STM has continued on a much larger scale than in the previous years, but little is known about the structure of collaboration networks that have contributed to its development and diversification.

## Social network theories and measures

A social network is a web of relationships connecting different actors together (e.g., individuals, organisations, nations). The purpose of analyzing networks in scientific research is to evaluate the performance of certain research actors through the structure and patterns of their relationships, as well as to guide research funding and development of science [76]. Following previous works [52, 77], SNA can be conducted through a variety of metrics such as ego-density at the network level; degree, betweenness and closeness centrality, efficiency and constraint at the actor level; and tie strength at the tie level [78, 79].

At the network level, density is the most basic network concept which measures the widespread of connectivity throughout the network as a whole [80]. In other words, it explains the extent of social activity in a network by determining the percentage of ties present [81]. On the other hand, ego-network density is used to describe the extent of connectivity in an ego's surrounding neighborhood [82]. In this study, the ego is either an author, institution or country. A dense network allows the dissemination of information throughout the network [83] and reflects a trustworthy environment for different actors [84]. However, a dense network is a two-edged sword where it might obstruct the ability of actors to access novel information outside their closely knitted cliques.

Actor level analysis was first pioneered through the "Bavelas–Leavitt Experiment" which involved five groups of undergraduate students, each had to communicate using a specific network structure (i.e. visualized as a 'star', 'Y', 'circle') to solve puzzles [85, 86]. It was found that the efficiency of information flow between group members was the highest in the centralized structures ('star' and 'Y'), leading to the formation of the network 'centrality' concept. Accordingly, Freeman [87] identified three measures of centrality which are degree, betweenness and closeness. Degree centrality that denotes the number of relationships a focal node has in the network. In other words, it is the number of co-authors associated with a given author. Degree centrality is mostly considered as a measure of 'immediate influence' or the ability of a node to directly affect others [88, 89]. Betweenness centrality is the number of shortest paths (between all pairs of nodes) that pass through a certain node [52]. Betweenness centrality is a good estimate of power and influence a node can exert on the resource flow between other actors [87, 90, 91]. A node with high betweenness centrality can be considered as an actor that regularly plays a bridging role among other actors in a network. On the other hand, closeness centrality measures the distance between a node and others in a network and reflects the speed in which information is spread across the entire network [87]. An actor with high closeness centrality is considered independent and can easily reach other actors without relying on intermediaries [81].

Another important actor level theory is Burt's [92] structural hole theory which highlights the importance of having holes (absence of ties) between actors to prevent redundant information. Otherwise, an actor can have redundant relationships by being connected to actors that themselves are connected, where maintaining these relationships could be costly and time consuming in which might constrain the performance of network actors. Burt proposed using 'efficiency' and 'constraint' to represent the presence of structural holes and redundant relationships, respectively.

Regarding tie level analysis, Granovetter [93] introduced the 'strength of weak ties' theory. He argued that individuals with weak relationships can obtain information at a faster rate than those with strong relationships. This is because individuals who are strongly connected to each other tend to share information most likely within their closely knitted clique than to transfer it to outsiders. In contrast, Krackhardt et al. [94] stressed on the importance of 'strong ties' to create a trustworthy environment, facilitate change and accelerate task completion. Additionally, Hansen [95] showed that strong ties rather than week ties can enhance the delivery of complex information.

## Materials and methods

### Data collection

This paper used co-authorship information to explore collaborative networks. The 'Web of Science' database was utilized with the search being restricted to journal articles with strings of ["stakeholder management" or "stakeholder analysis" or "stakeholder identification" or "stakeholder theory" or "stakeholder engagement" or "stakeholder influence"] in their title, abstract or keywords. These are the most frequently used themes in stakeholder research to describe the concept of STM. Other types of documents such as conference proceedings, and books were excluded. The year 1989 was chosen as the outset date of our research because the results of Laplume et al. [96] and the web of science search showed that the first stakeholder-based scientific article was published in 1989.

In order to have a better understanding of the evolution of collaboration networks, different datasets were required. Therefore, the overall time period of 32 years was split into three consecutive sub-periods, that being $t_1$: 1989–1999), $t_2$: 2000–2010 and $t_3$: 2011–2020. The bibliometric data for each phase was extracted independently in plain text format (compatible with Bibexcel package program for bibliometric analysis) and involved manuscript titles, authors' names and affiliations, journal titles, institutional names, identification numbers, abstracts, keywords, publication dates, etc. Out of 21,173 authors, 3115 were duplicates, so 19,058 authors were sent for further analysis. The number of articles extracted was 85 for $t_1$, 885 for $t_2$ and 5157 for $t_3$, counting for a total number of 6127 articles.

### Data refinement

The bibliometric datasets for the three sub-periods were imported into Bibexcel package program [97] for data preparation and co-occurrence analysis. Fig 1 summarizes the entire methodological process used for extracting and analyzing the data. The first issue encountered was to resolve name authority control problems (i.e. different entities with same names, or same entities with different names [27]. For instance, some journal articles were the same but had different titles (e.g., 'Moving beyond dyadic ties' and 'Moving beyond Dyadic Ties: A Network Theory of Stakeholder Influences'). Therefore, a standardization process was conducted by removing duplicates (i.e., articles with same DOI were considered as one source). Moreover, it was important to convert upper and lower cases (e.g., WICKS AC, Wicks AC) of all records to a standard lower-case format (Wicks AC) to avoid duplication of records that might impact

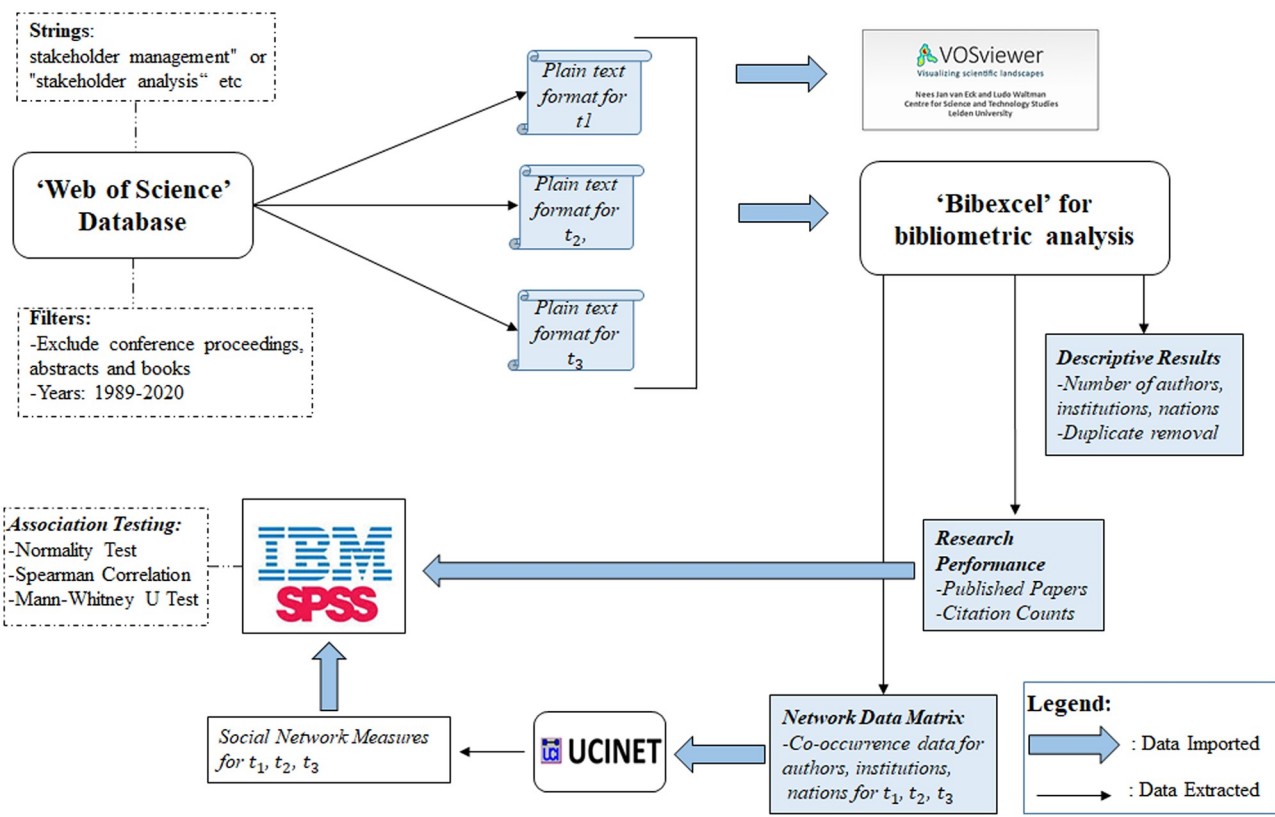

**Fig 1. Methodological process for extracting and analyzing bibliometric and social network data.**

network structure. For some of the records, especially that of institutions and countries, it has been shown that co-occurrence has occurred between the same institutions and the same countries. In this case, the names were not brought together but kept apart due to the fact that collaboration has happened between authors of the same institution, or between institutions of the same country. In other words, self-loops were not excluded from our analysis. Using Bibexcel, we extracted social network data for each of the authors, institutions and countries networks and for each sub-period, that involved information about the presence and absence of relationships between the actors. Then, the data was imported into excel and manually scrutinized to correct possible spelling errors.

## Social network analysis

The matrices were imported into an SNA program used by many network scholars—"UCINET 6.0" [98] to calculate the social network measures for each matrix. UCINET is a SNA software mainly used for whole network studies, which features a large number of network metrics to quantify patterns of relationships. Centrality measures were calculated for the authors, institutions and countries to determine those that are leading collaborative works in their networks. However, further network measures such as ego-density, efficiency, constraint and average tie strength were only calculated for the authors to cohesively understand the longitudinal effect of co-authorship networks on research performance.

Ego-density, degree centrality, betweenness centrality and closeness centrality, efficiency and constraint were calculated for each author, institution and country and for each sub-period.

Ego-density is number of actual ties not involving the ego divided by the number of possible ties in an ego network:

$$ED = \frac{\sum_{i,\,j=1}^{n} Z_{ij}}{n(n-1)/2}$$

where n refers to the number of alters the ego is connected to, $Z_{ij}$ is the tie strength between actors i and j and (n (n − 1))⁄2 refers to hightest possible number of ties.

Degree centrality is the count of contacts a focal node has in a network [99]. It is not reasonable to compare a node with a centrality score of 20 in a network of 50 nodes with a node of same centrality score but in a smaller network of 15 nodes. Therefore, in order to understand the extent to which authors are central in a network and compare their centrality across different networks that vary in size, Freeman's [100] normalized measures (n-1) for degree, betweenness and closeness centrality are used. Normalized degree centrality:

$$\frac{D_i = \sum_j z_{ij}}{n-1}$$

Where $i$ is the focal node, $j$ is any other actor and $z_{ij} = 1$ for an existing tie between $i$ and $j$.

Normalized betweenness centrality is calculated as the proportional number of times a focal node lies on the shortest path between other actors [101]:

$$B_i = \sum \frac{\frac{z_{jq}(i)}{z_{jq}}}{n-1}$$

where $i$ is the focal node, $j$ and $q$ are any other two actors, $z_{jq}$ is the total number of shortest paths from $j$ to $q$, and $z_{jq}(i)$ is the contribution of $i$ to those paths.

Normalized closeness centrality is the total number of distances between the focal node and all other nodes:

$$C_i = \sum \frac{z(pj, pq)^{-1}}{n-1}$$

where $z(pj, pq)$ is the shortest distance between node $pj$ and node $pq$ in the network.

Efficiency is measured by dividing the number of non-redundant actors divided by network size:

$$E_i = \frac{\sum \left[1 - \sum p_{iq} m_{jq}\right], \; q \neq i, j}{N}$$

where $i$ is the focal node, $j$ and $q$ are any other two actors, $p_{iq}$ is the tie strength between i and j and $m_{jq}$ is the tie strength between j and q. N is the number of alters in the ego network.

Conversely, network constraint measures the extent to which an actor's time and energy are invested in contacts who are themselves are connected to one another [102]:

$$\left(p_{ij} + \sum_q p_{iq} p_{qj}\right)^2, q \neq i, j$$

where i is the ego having a strong tie with j (represented by $p_{ij}$), j is another alter having a strong tie with I (reprenseted by $p_{iq}$) and q is also an another alter having a strong tie with j (represented by $p_{qj}$).

Mean tie strength is the sum of the strength of all ties of an ego (outgoing and ingoing), each tie strength ranging from 1 to 4, divided by the number of alters in a network:

$$T_i = \frac{\sum j\, S_{qj}}{N_q}$$

where $j$ is the ego, q is the alter, $S_{qj}$ is the tie strength between j and q, and $N$q is the number of alters in an ego's network.

### Sociograms

To construct and visualize the collaboration networks of authors, institutions and countries, bibliometric data from WoS was directly imported into VOSviewer–a specialized software tool that visualizes networks based on scientific publications [103].

### Data analysis

To understand the association between social network measures and research performance, the extracted social network measures from UCINET were imported into SPSS with the number of citations and documents published for each author. Correlation and T-tests determined whether a positive or a negative association exists between the explored variables.

## Results and discussion

### Descriptive results

A total of 6127 articles were obtained from different journals between 1989 and 2020. As seen in Table 1 and Fig 2, there is an exponential increase in the number of published articles. 85 articles were published in $t_1$, 885 in $t_2$ and remarkably 5157 in $t_3$. This shows that the majority of collaborative endeavors have occurred in the last decade with a 482% increase in the number of articles from $t_2$ to $t_3$. The number of articles written by multi authors (three or more authors) in the last 32 years is 3590 (58.5%) which is much higher than double author articles (1603 articles, 26.16%) and single author articles (934 papers, 16.2%). Fig 2 shows that the

**Table 1. Descriptive results of scientific collaboration and network properties in STM.**

|  | $t_1$: 1989–1999 | $t_2$: 2000–2010 | $t_3$: 2011–2020 |
|---|---|---|---|
| Number of articles | 85 | 885 | 5157 |
| Singe-authors articles | 37 | 241 | 656 |
| Double-author articles | 28 | 306 | 1269 |
| Multi-author articles | 20 | 338 | 3232 |
| Number of Authors | 156 | 1997 | 16905 |
| Total number of citations | 19476 | 25052 | 61942 |
| Number of Institutions | 88 | 879 | 3778 |
| Number of Countries | 16 | 74 | 141 |
| **Network Statistics** | | | |
| Number of Cliques (3 and more actor) | 19 | 88 | 232 |
| Size of largest clique | 4 nodes | 12 nodes | 31 nodes |
| Author Network Density | 0.018 | 0.004 | 0.010 |
| Institution Network Density | 0.010 | 0.003 | 0.014 |
| Country Network Density | 0.067 | 0.112 | 0.1 |
| Degree Centralization | 0.040 | 0.025 | 0.037 |

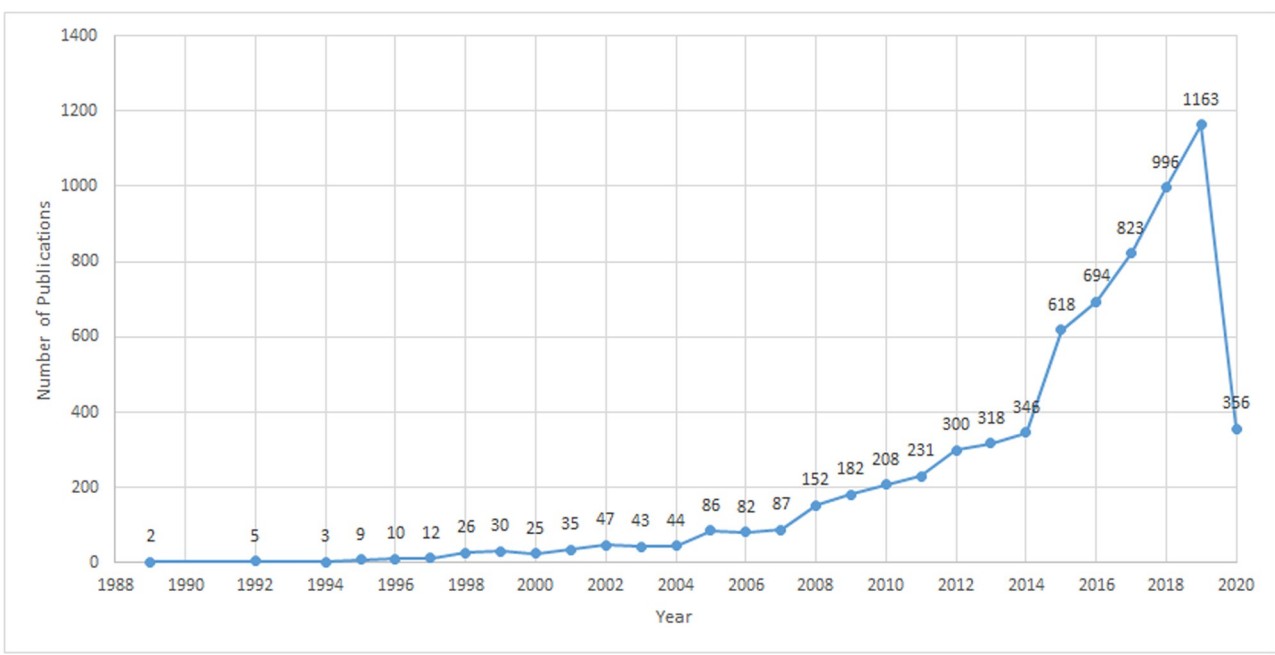

**Fig 2. Number of journal articles published per year in STM discipline.**

number of published articles increased gradually from 2 to 44 articles between 1989 and 2004, with an exponential increase in 2005 and onwards (i.e., the number of publications in 2004 has been doubled in 2005). The period from 2014 and 2019 experienced the highest number of published articles, indicating the increased interest of the academic community in STM research.

Regarding institutional co-occurrence, it is evident that $t_3$ has witnessed the highest number of collaborative institutions (3778) than $t_2$ (879) and $t_1$ (132). Similarly, the number of collaborating countries was the highest (155) in $t_3$ and the lowest (16) in $t_1$. Given that a scientific field might require 45 years to mature [104], the overall results show that the STM field moved from incubation ($t_1$) to incremental growth ($t_2$) to maturity ($t_3$), reflected by the dramatic increase in the number of articles, institutions, countries and in the number of citations (106,466 in total) especially in $t_3$ (61,942).

## Social network analysis results

Using SNA, the 10 most prolific and influential actors for each network (authors, institutions, countries) in each sub-period ($t_1$, $t_2$, $t_3$) were identified.

**Authors.** Table 2 shows that Bair JD is considered the most prolific author in $t_1$ with the most direct connections (degree centrality = 0.045) (all centrality measures are normalized) and the largest betweenness centrality ($8^{10^{-3}}$) and is considered the closest to all other actors in the network (closeness centrality = 0.343). Bosse GC, Driskill JM and Fottler MD are next in line with same centrality scores, followed by Friedman R, Jones TM, Berman SL, Agle BR and Sonnenfeld JA. Fig 3 shows the evolution of collaborative networks of co-authors by sub-period. Surprisingly, it is shown that some of these authors share the same clique, especially for Bair JD, Bosse GC and Driskill JM, but the majority of the authors in Table 2 do not belong to a single integral clique.

**Table 2. Ranking of 10 most prolific authors based on their centrality measures for each sub-period.**

| $t_1$: 1989–1999 | | | | $t_2$: 2000–2010 | | | |
|---|---|---|---|---|---|---|---|
| Author | Degree | Betweennes | Closeness | Author | Degree | Betweenness | Closeness |
| Blair JD | 0.045 | $8^{10^{-3}}$ | 0.343 | Boitani I | 0.007 | 0 | 0.201 |
| Bosse GC | 0.032 | 0 | 0.342 | Turner W | 0.007 | 0 | 0.201 |
| Driskill JM | 0.032 | 0 | 0.342 | Barnett J | 0.006 | 0 | 0.200 |
| Fottler MD | 0.032 | 0 | 0.342 | Brown K | 0.006 | 0 | 0.200 |
| Baker CM | 0.032 | 0 | 0.340 | Freeeman RE | 0.004 | $1.22^{10^{-5}}$ | 0.201 |
| Friedman R | 0.025 | 0 | 0.339 | Grant T | 0.004 | $1.2^{10^{-5}}$ | 0.201 |
| Jones TM | 0.025 | 0 | 0.339 | Bloom G | 0.005 | 0 | 0.200 |
| Berman SL | 0.019 | 0 | 0.338 | Berrone P | 0.003 | $1.5^{10^{-5}}$ | 0.200 |
| Agle BR | 0.019 | 0 | 0.337 | Robert A | 0.002 | $1^{10^{-5}}$ | 0.200 |
| Sonnenfeld JA | 0.012 | 0 | 0.336 | Andersson I | 0.001 | 0 | 0.200 |
| $t_3$: 2011–2020 | | | | | | | |
| Author | | Degree | | Betweenness | | Closeness | |
| Tugwell P | | 0.014 | | $5^{10^{-3}}$ | | 0.078 | |
| Graham ID | | 0.013 | | $2^{10^{-3}}$ | | 0.078 | |
| Newman PA | | 0.014 | | $1^{10^{-4}}$ | | 0.072 | |
| Dawkins JS | | 0.015 | | $4.1^{10^{-5}}$ | | 0.072 | |
| Walker CE | | 0.015 | | $4.1^{10^{-5}}$ | | 0.072 | |
| Cancannon TW | | 0.012 | | $1^{10^{-3}}$ | | 0.078 | |
| Tucker J | | 0.012 | | $1^{10^{-4}}$ | | 0.072 | |
| Guise JM | | 0.011 | | $8^{10^{-4}}$ | | 0.077 | |
| Crowe S | | 0.011 | | $3^{10^{-4}}$ | | 0.077 | |
| Grant S | | 0.010 | | $5^{10^{-4}}$ | | 0.077 | |

This indicates that collaboration is in the form of sub-networks of closely knitted authors each forming their own collaborative clique. It is evident that collaboration is still premature with only 156 authors not well connected in the network. $t_1$ is known as the discovery period of stakeholder theory where it first appeared in management journals (e.g. Academy of Management Review) [32].

In $t_2$, the collaboration network consists of 1957 authors and has become larger and more condensed than in $t_1$. However, it is important to note that Table 1 earlier shows that 62% of articles (547 out of 885 articles) are single and double authored and only 38% (338 articles) are multi-authored. This finding can be noted in Network B, Fig 3 with the emergence of more than 1000 single and dyadic authors that have further fragmented the collaboration network as a whole. This disintegration of the stakeholder domain is expected because the stakeholder theory has a wide scope of interpretations and the term 'stakeholder' can mean different things to different people [105]. With the increase in stakeholder theoretical disputes between the moral justifications [41] and managerial implications of the theory [38, 66, 105], numerous solo, dyadic and triadic have risen, detaching from both the mainstream stakeholder theory research [34, 35], and the large network cliques [106, 107]. Perhaps, a reason why most of the prolific actors in $t_1$ did not make the list in $t_2$ is because new research areas have emerged, such as stakeholder engagement [108, 109], stakeholder social network analysis [56, 110], stakeholder involvement in policy decision making [111] and many more.

(A)

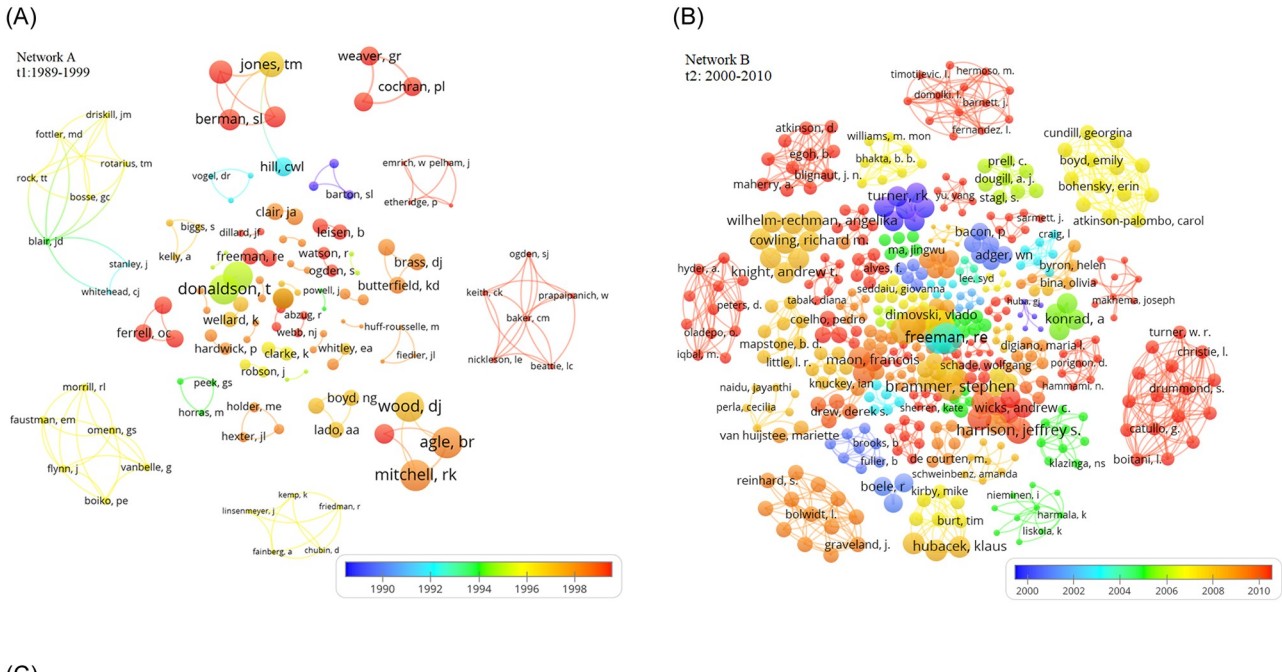

(C)

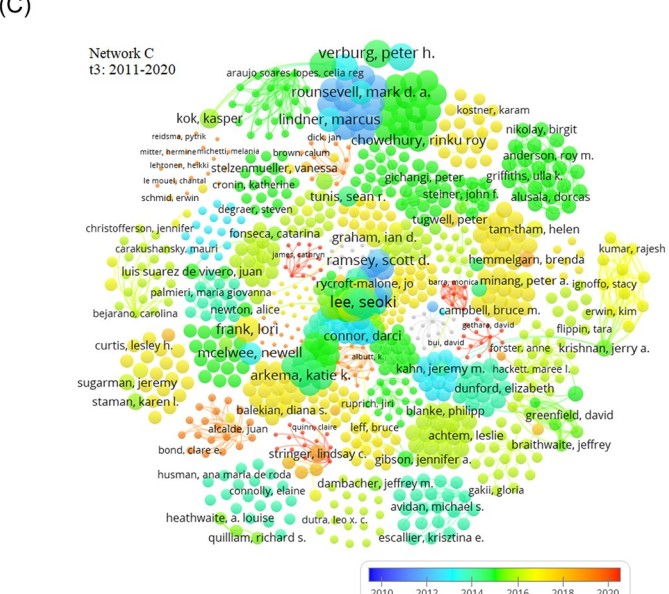

**Fig 3. Co-authorship networks in $t_1$ (Network A), $t_2$ (Network B) and $t_3$ (Network C).** Each node/circle represents a researcher who have published in the STM field. The size of each node size is proportional to the number of citations. A line connecting two nodes indicates an, at least, one published paper between two authors in STM field.

Larger cliques are observed which some reaching to 16 authors and with the emergence of numerous small to medium size sub-networks. For $t_2$, a totally new set of influential authors have emerged but being less central than those in $t_1$ with lower degree and closeness centrality scores but with higher betweenness in general. This indicates that collaboration endeavors are mainly driven by clique members rather than by highly central actors. Similarly, another study showed that key authors are more likely to form a well-connected group that collaborates frequently and diversely [112], rather to collaborate solely through central actors. Among the

most influential actors are Boitani I and Turner W who have the same centrality scores, followed by Barnett J, Brown K, then Freeman RE and Grant T who have a lower degree centrality (0.004) but are still considered highly central by occupying a strong brokerage position (betweenness centrality is $1.22^{10^{-5}}$ and $1.2^{10^{-5}}$ respectively). Bloom G, Berron P, Robert A and Andersson I are less central but still considered highly influential.

As it can be interpreted from the graphical visualization in Fig 3, that the scenario observed in $t_3$ is very similar to that in $t_2$, but with a larger network of 16,905 authors (763% increase in number of authors from $t_2$). In particular, the number of components has increased to 88 and expanded to include 12 actors. In contrast, network density–the percentage of existing ties over the total number of possible ties–has decreased from 1.8% in $t_1$ to 0.08% in $t_3$. Although it seems intuitive that density would increase with new researchers entering the field, this did not seem to be the case where density decreased with further fragmentations that reduced the number of connections as the number of nodes increased. This finding is supported by a study [18] that found a decrease in network density of author collaboration networks from 0.026 in the 1980s to 0.003 in the 2000s. In the presence of 16905 authors with different research interests, it is nearly impossible to connect the majority of the nodes and achieve a high network density. The overlay color range in Network C, Fig 3 also shows that the majority of publications have occurred between 2014 and 2018 with few co-authorships noted in the last two years.

The SNA results presented in Table 2 show that Tugwell P is the most influential author in the network, followed by Graham ID, Newman PA, Dawkins JS and Walker CE who all have higher degree centrality scores than the rest. Remarkably, the findings of betweenness centrality in $t_3$ show an increase in the importance of the intermediary role, as all prominent actors (see Table 2) have a higher betweenness centrality score compared to that of $t_1$ and $t_3$. The brokerage role is significant in $t_3$ with the decrease in degree and closeness scores. Therefore, the collaboration network has become more dependent on authors with a brokerage role in $t_3$.

The evolution of the collaboration network across three decades shows that the STM authors do not belong to the same network. This observation has also been reported in the Network Meta-Analysis field where collaborating authors belonged to different network clusters [113]. Therefore, the collaboration network can be best described as involving a high number of authors with different research interests that have pursued different research areas by either being a part of a sub-network of three or more actors or by working alone or in pairs. Evidence for radical changes in network structures from $t_1$ to $t_3$, other than the increase in component sizes and fragmentation, have not been demonstrated, where this is still considered an important and unexpected finding. The findings show that the stakeholder concept is a multidisciplinary theory applied in various research domains such as in health care management [114–119], marine policy [120, 121], agriculture [24, 122], applied geography [123, 124], engineering and architecture [23, 125], marketing [126–128], public affairs [25, 129–131], project management [73, 132–134] and tourism [135–137]. In other words, the stakeholder concept has been developed mainly by multidisciplinary teams of both experienced and emerging scientists. Therefore, this finding contradicts what has been recently speculated that STM is still at an early stage and that published studies are still limited [138].

**Institutions.** Institutional collaboration enables the sharing of unique resources and improves research visibility and contribution [16]. The results show that the first period contained 88 institutions that have participated in stakeholder research. Surprisingly, 8 out of the 10 most collaborative central institutions are from the United States (see Table 3) and are Health Management Link (Indianapolis, USA), Indiana University, University of Iowa, Kings Daughters Hospital, Penn State University, Washington State University, Colorado State

**Table 3. Ranking of 10 most prolific institutions based on their centrality measures for each sub-period.**

| $t_1$: 1989–1999 | | | | $t_2$: 2000–2010 | | | |
|---|---|---|---|---|---|---|---|
| Institution | Degree | Betweenness | Closeness | Institution | Degree | Betweenness | Closeness |
| Hlth Mgt Link | 0.046 | 0 | 0.343 | Erasmus Uni | 0.028 | 0.050 | 0.100 |
| Indiana Uni | 0.046 | 0 | 0.343 | York Univ | 0.020 | 0.050 | 0.100 |
| Uni Iowa | 0.046 | 0 | 0.343 | Uni London | 0.020 | 0.038 | 0.100 |
| Kings Daughter Hosp | 0.046 | 0 | 0.343 | Uni Queensland | 0.020 | 0.022 | 0.090 |
| Penn State Univ | 0.034 | $5.3^{10^{-4}}$ | 0.341 | Uni East Anglia | 0.020 | 0.014 | 0.098 |
| Washington State Uni | 0.023 | 0 | 0.339 | Uni North Carolina | 0.018 | 0.024 | 0.096 |
| Colorado State Uni | 0.023 | 0 | 0.338 | Harvard Uni | 0.018 | 0.017 | 0.096 |
| Uni Groningen | 0.023 | 0 | 0.338 | Uni Autonoma Barcelona | 0.018 | 0.016 | 0.094 |
| Boston Uni | 0.014 | 0 | 0.335 | Univ Utrecht | 0.018 | 0.012 | 0.100 |
| Bournemouth Uni | 0.014 | 0 | 0.335 | Uni Verona | 0.017 | 0 | 0.09 |

| $t_3$: 2011–2020 | | | |
|---|---|---|---|
| Institution | Degree | Betweenness | Closeness |
| Uni Leeds | 0.100 | 0.067 | 0.433 |
| Univ Toronto | 0.099 | 0.053 | 0.431 |
| Univ Washington | 0.092 | 0.072 | 0.423 |
| Univ Calgary | 0.083 | 0.048 | 0.400 |
| Univ Ottawa | 0.080 | 0.033 | 0.419 |
| Univ Oxford | 0.075 | 0.053 | 0.426 |
| Univ British Columbia | 0.073 | 0.041 | 0.400 |
| Univ Melbourne | 0.073 | 0.028 | 0.404 |
| Univ Sydney | 0.071 | 0.029 | 0.399 |
| Harvard Univ | 0.069 | 0.041 | 0.400 |

University and Boston University. Similar to the author collaboration network in $t_1$ (Network A, Fig 3), the institutional network (Network A, Fig 4) shows that the collaboration network doesn't constitute a main component but is disseminated into several small size components (3 to 5 nodes). This shows that the above institutions are only influential in their own cliques.

In contrast to $t_1$, $t_2$ has witnessed a wider international collaboration where 8 out of the 10 most prolific institutions are from outside the US (see Table 3), also being the top 5 institutions and are Erasmus University (Netherlands) which has the highest degree centrality (0.028) and being the most influential intermediary with York University (Canada) (Betweenness centrality = 0.05), University of London (UK), University of Queensland (Australia), University of East Anglia (UK); followed by two US institutions–University of North Carolina and Harvard University, and then Autonomous University of Barcelona (Spain), Utrecht University (Netherlands) and Aarhus University (Denmark). This result is interestingly surprising as it challenges previous studies that showed that most published papers, in general, are from USA, UK and Canada, which also are the most central in collaboration networks [1, 16, 139].

Regarding the network structure and contrary to the institutional network in $t_1$, the result show the emergence of a main component in $t_2$ that is well connected and highly centralized by constituting a nucleus of all of the above prolific institutions, but surrounded by numerous institutions that are isolates (i.e. nodes disconnected from the main component). However, a deeper inspection reveals that an institution can also be considered highly influential without being embedded in the main component, such as in the case of Autonomous University of Barcelona (placed between the main component and the isolates in Fig 4, Network B).

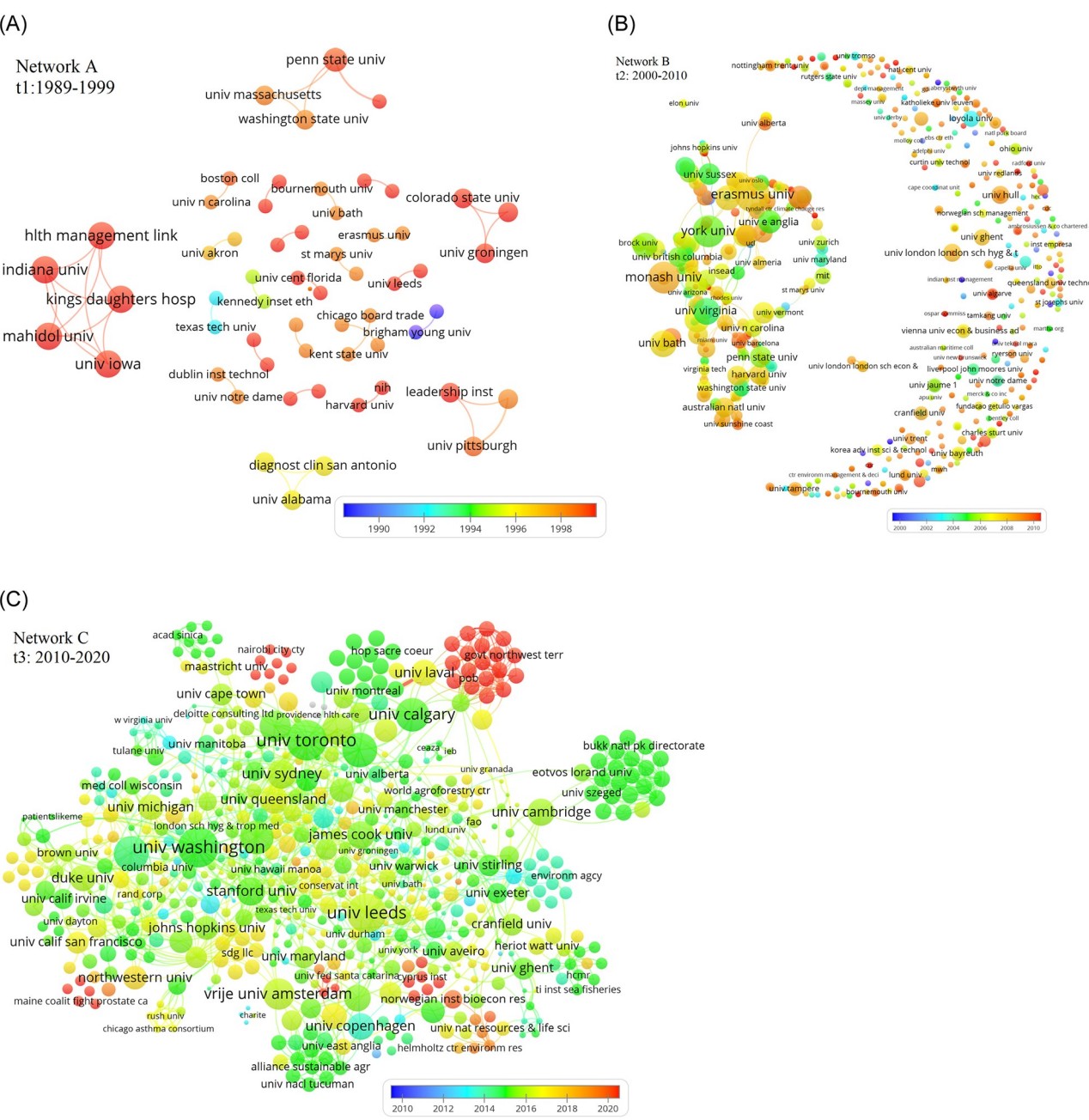

**Fig 4. Co-occurrence networks of institutions in $t_1$ (Network A), $t_2$ (Network B) and $t_3$ (Network C).** Each node/circle represents an institution that has been involved in STM research. The size of each node size is proportional to the number of connections. A line connecting two nodes indicates a collaborative relationship between two institutions in the STM field.

Autonomous University of Barcelona is connected to 16 other institutions present in its own clique, such as Queen Mary University of London, Medical University of Vienna and Illinois state university. This analysis reinforces the important role of cliques in facilitating collaborating processes. The findings overall place STM research on the global radar by being in favor of the most prestige universities worldwide such as University of London, Harvard University and University of Queensland.

The results for $t_3$ show University of Leeds being the most prominent institution with the highest degree, betweenness and closeness centrality, followed by the University of Toronto, University of Washington, University of Calgary, University of Oxford, University of Otawa, University of Oxford, University of British Colombia, University of Melbourne, University of Sydney and Harvard University. Most of these institutions do not belong to the same components and therefore, it can be argued that collaboration is led by highly central actors dissemininated across the entire network. This has facilitated the connection of detached neighbourhoods as reflected by the increase in density from 0.003 in period 2 to 0.014 in period 3 (367% increase in density). This finding is contrary to Koseoglu [20] who found that collaboration network density in strategic management research did not increase across 34 years despite the increase in network size.

For this reason, each period is characterised by having a very distinct list of prolific actors that change with the change in network size and structure. Moreover, the number of vertices has dramatically increased from 1201 in period 2 (879 nodes) to 12833 in period 3 (3778 nodes). It can be argued that interesting patterns were observed in the institutional network for $t_3$, especially with the reduction of isolates, the increased density and the enlargement of the main component in $t_2$ to include other large cliques that reached 31 nodes (158% increase in clique size). This finding contradicts previous research in strategic management that showed that large institutional cliques did not emerge with the enlargement of collaboration network [20]. The overlay color range in Network C in Fig 4 shows that the majority of institutions have published between 2014 and 2018 with a continual rise in 2019 and 2020.

**Countries.** Table 4 provides interesting observations where USA and England are the most prolific actors that are leading collaborative research in the last 32 years. This finding is also supported by previous studies that showed that countries in North and South America, with Europe, are the best-connected countries to facilate international research collaboration [20, 139–141]. The collaboration network in $t_1$ only exists because of the brokerage roles performed by USA and England (see Network A, Fig 5). USA stands out by having the most direct relationships (degree centrality = 0.4), brokerage position (betweenness centrality = 0.142) and being the closest to all other actors (closeness centrality = 0.454). USA and England are considered 'cutpoints' that if removed would disconnect the entire two networks. For this reason, the rest of the countries (Australia, Canada, Scotland, etc) are considered prolific only because of their only single relationship with either USA or England. A number of isolates are also noted and are Wales, Israel, Belgium, Sweden, Spain and New Zealand.

Unlike the scenario in $t_1$, a significant involvement of new countries in the collaboration network is observed in $t_2$ while still having USA and England as the most central actors. An interesting finding is that the majority of countries that followed USA and England were not among the prolific actors in $t_1$, such as Germany, Italy, Belgium, Spain and Denmark. On other hand, some countries that existed in $t_1$, such as Australia, Cananda and Netherlands, have taken a more significant role in the collaboration network in $t_2$, while Scotland, Hungary, Thailand, Jamaica and Ireland have dissappeared from the prolific radar for $t_2$ and $t_3$. Remarkable, the network density of the country contribution network in $t_2$ and $t_3$ are 11.2% and 10% which are considered the highest compared to all of the previous networks in most decades. Fig 5 shows that the collaboration network of countries started by being uncondensed, fragmented and highly centralised with 16 countries controlling the marjority of connections, to a highly dense, less fragmented network of 74 countries in $t_2$, to a larger network of 141 countries and 1059 vertices counting for a 10% density in $t_3$. Network 3, Fig 5 shows that the majority of countries emerged between 2014 and 2017.

To our knowledge, a well connected network of collaborative countries as observed in $t_2$ and $t_3$ is not occasional. Geographic, linguistic and cultural distances between scientists of

**Table 4. Ranking of 10 most prolific countries based on their centrality measures for each sub-period.**

| $t_1$: 1989–1999 | | | | $t_2$: 2000–2010 | | | |
|---|---|---|---|---|---|---|---|
| Country | Degree | Betweenness | Closeness | Country | Degree | Betweenness | Closeness |
| USA | 0.400 | 0.142 | 0.454 | England | 0.647 | 0.311 | 0.606 |
| England | 0.133 | 0.009 | 0.365 | USA | 0.535 | 0.228 | 0.563 |
| Australia | 0.066 | 0 | 0.394 | Germany | 0.338 | 0.034 | 0.5036 |
| Canada | 0.066 | 0 | 0.394 | Italy | 0.309 | 0.029 | 0.503 |
| Ireland | 0.066 | 0 | 0.394 | Netherlands | 0.281 | 0.044 | 0.486 |
| Jamaica | 0.066 | 0 | 0.394 | Belgium | 0.295 | 0.025 | 0.489 |
| Netherlands | 0.066 | 0 | 0.394 | Spain | 0.281 | 0.017 | 0.486 |
| Thailand | 0.066 | 0 | 0.394 | Australia | 0.253 | 0.036 | 0.482 |
| Hungary | 0.066 | 0 | 0.357 | Denmark | 0.267 | 0.020 | 0.479 |
| Scotland | 0.066 | 0 | 0.357 | Canada | 0.253 | 0.016 | 0.479 |

| $t_3$: 2011–2020 | | | |
|---|---|---|---|
| Country | Degree | Betweenness | Closeness |
| USA | 0.624 | 0.212 | 0.665 |
| England | 0.595 | 0.141 | 0.652 |
| Netherlands | 0.489 | 0.092 | 0.607 |
| Canada | 0.453 | 0.051 | 0.597 |
| Australia | 0.432 | 0.063 | 0.587 |
| France | 0.375 | 0.046 | 0.559 |
| Germany | 0.347 | 0.056 | 0.548 |
| Spain | 0.361 | 0.024 | 0.550 |
| Switzerland | 0.347 | 0.022 | 0.559 |
| Belgium | 0.326 | 0.021 | 0.550 |

different countries may significantly impact collaboration prevalence [142, 143]. According to Li et al. (2016), it is more often for collaboration to occur within the same country or same institution due to many reasons including the ease of communication, low intra-competition and low funding opportunities. For example, a study on how higher educations perceive stakeholder salience was possible due to the collaboration of Benneworth and Jongbloed [144] who both were researchers at the University of Twente in the Netherlands. However, the findings in this study allowed us to observe cross country collaboration since the origin of stakeholder theory in the 1980s. Perhaps a contributing reason for this global collaboration, at least in part, is the presence of several funding agencies, such as the Economic and Social Research Council (ESRC), that supported many stakeholder research studies which brought together many scientist from different countries such as Wales, England, Spain and Sweden [145, 146].

### Effect of co-authorship networks on research productivity and citation-based performance

A preliminary investigation of the associations involved exploring the correlations between actors' network attributes and research performance for each period. Since the assumption of normality has been violated, non-parametric tests of Spearman correlation and Mann-Whitney U Test were conducted. The results in Table 5 show that the correlations varied differently across the three sub-periods with regards to magnitude, direction and significance. Research productivity is shown to have the strongest correlation with tie strength in $t_1$ (r = -0.39, p < 0.01), betweeness centrality in $t_2$ (r = 0.67, p < 0.01) and ego-density in $t_3$ (r = -0.563,

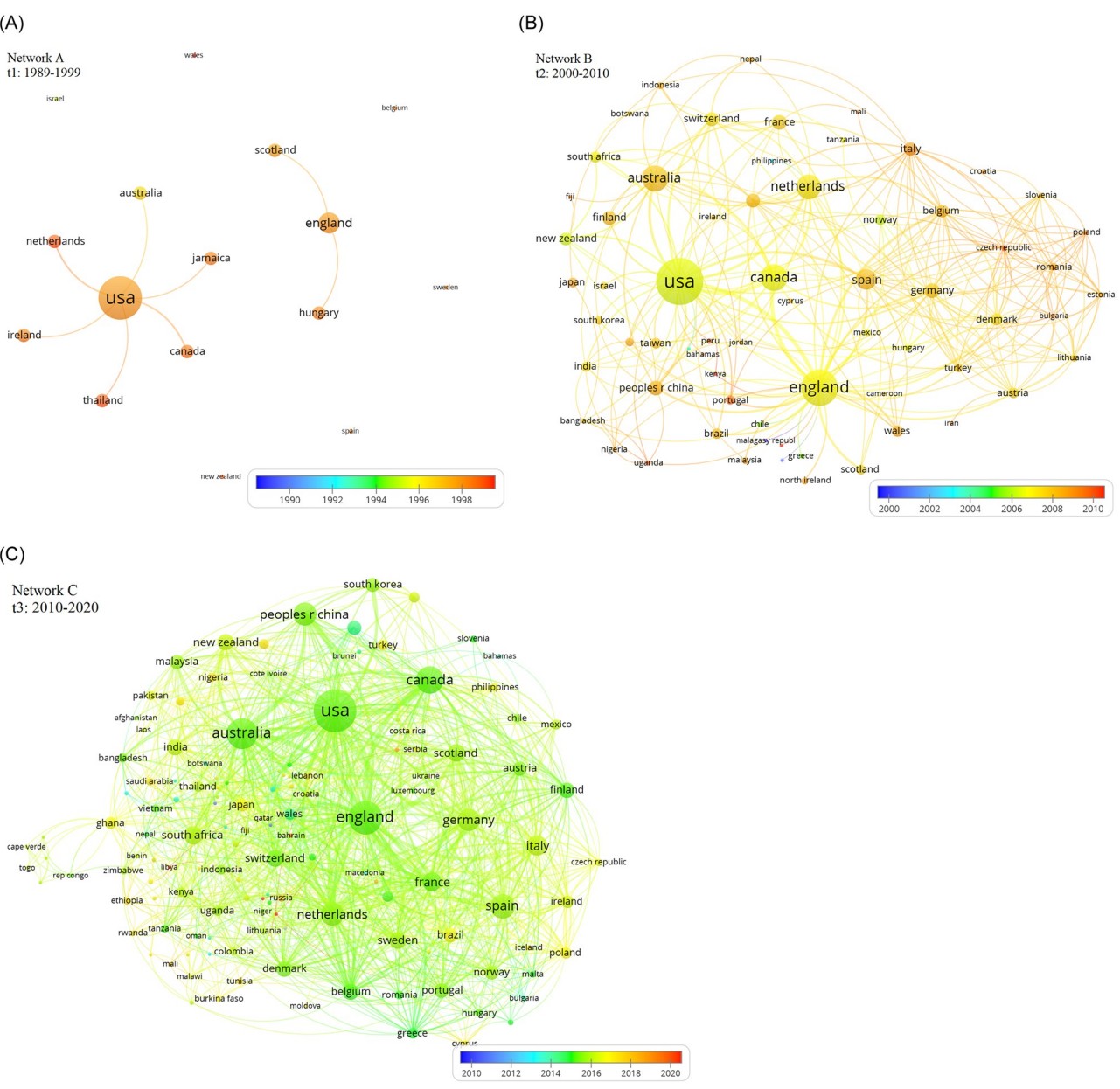

**Fig 5. Co-occurrence networks of countries in $t_1$ (Network A), $t_2$ (Network B) and $t_3$ (Network C).** Each node/circle represents a country that has been involved in STM research. The size of each node size is proportional to the number of connections. A line connecting two nodes indicates a collaborative relationship between two countries in the STM field.

p < 0.01). On the other hand, citation counts is mostly correlated with tie strength in $t_1$ (r = 0.49, p < 0.01) and $t_2$ (r = 0.48, p < 0.01).

Remarkably, the correlations between research productivity and each of degree centrality (r = -0.19, p < 0.01) and tie strength (r = -0.04, p < 0.01) in $t_3$, have shifted its direction as opposed to the positive correlations in $t_1$ and $t_2$. The results overall show that all social network variables (ego-density, betweenness, closeness, efficiency, contraint, tie strength) are either negatively or positively correlated with research performance (i.e., citation counts, research productivity) (see Table 5 for more information).

**Table 5. Spearman correlation test results between social network and research performance variables.**

| 1989–1999 | | 1 | 2 | 3 | 4 | 5 | 6 | 7 | 8 | 9 |
|---|---|---|---|---|---|---|---|---|---|---|
| | Ego-Density (1) | 1 | | | | | | | | |
| | Degree (2) | .66** | 1 | | | | | | | |
| | Betweenness (3) | .00 | .00 | 1 | | | | | | |
| | Closeness (4) | .97** | .68** | .00 | 1 | | | | | |
| | Efficiency (5) | -.17 | .32** | .00 | -.15 | 1 | | | | |
| | Constraint (6) | .96** | .53** | .00 | .94** | -.10 | 1 | | | |
| | Tie Strength (7) | .27** | .36** | .00 | -.39** | .10 | .20* | 1 | | |
| | Citations (8) | .00 | -.038 | .00 | -.015 | .18* | .04 | .49** | 1 | |
| | Productivity (9) | .200* | .35** | .00 | .226** | .27 | .19* | -.39* | .30** | 1 |
| 2000–2010 | Ego-Density (1) | 1 | | | | | | | | |
| | Degree (2) | .81** | 1 | | | | | | | |
| | Betweenness (3) | -.08** | .16** | 1 | | | | | | |
| | Closeness (4) | .78** | .96** | .18** | 1 | | | | | |
| | Efficiency (5) | -.40** | -.40** | .12** | -.36** | 1 | | | | |
| | Constraint (6) | .86** | .51** | .00 | .5** | -.14** | 1 | | | |
| | Tie Strength (7) | -.06** | -.08 | .22** | .03 | .17** | -.03 | 1 | | |
| | Citations (8) | 08** | 09* | .19** | .11** | .09** | .08** | .48** | 1 | |
| | Productivity (9) | -.03 | .07** | .67** | .09** | .10** | -.02 | .32** | .27** | 1 |
| 2011–2020 | Ego-Density (1) | 1 | | | | | | | | |
| | Degree (2) | -.13** | 1 | | | | | | | |
| | Betweenness (3) | -.7** | .17** | 1 | | | | | | |
| | Closeness (4) | -.17** | .77** | .19** | 1 | | | | | |
| | Efficiency (5) | -.36** | -.76** | .25** | -.49** | 1 | | | | |
| | Constraint (6) | .15** | -.84** | -.17** | -.62** | 68** | 1 | | | |
| | Tie Strength (7) | .00 | .93** | .21** | -.73** | -.00 | -.00 | 1 | | |
| | Citations (8) | -.25** | .07** | .16** | .12* | .10** | -.15** | .11** | 1 | |
| | Productivity (9) | -.63** | -.19** | .39** | -.12** | .46** | -.03* | -.04** | .37** | 1 |

To explore the association between ego-density and research performance, the median for ego-density index was chosen as a cut point to segregate the participants into two groups: authors with ego-density scores above the median and are considered as "high ego-density group" and authors with ego-density scores lower than the median and are considered as "low ego-density group". The results of the Mann-Whitney U test (U = 2658, z = -2.86, p = 0.04) summarized in Table 6 show a positive association in $t_1$ with higher research performance scores observed in the high-density group (Mdn = 83) than the low density group (Mdn = 75). Similarly, the results (U = 443079, z = -6.6, p = 0.00) show a positive association in $t_2$ with higher research performance scores observed in the high density group (Mdn = 1015) than the low density group (Mdn = 973). Accordingly, we argue that it was essential to have highly dense collaborative clusters in the first decade to publish scientific papers that can bring awareness to stakeholder theory as a newly developed theory of management and ethics.

The results show that degree centrality is positively associated with both research productivity andcitation counts in $t_2$ while no association in $t_1$. In particular, authors with numerous collaborative relationships in $t_2$ had higher citation counts (Mdn = 1042) and research productivity (Mdn = 1011) than those with fewer relationships (Mdn = 925 and Mdn = 977 respectively); U = 411370, p = 0.00 and U = 449944, p = 0.03 respectively. In $t_3$, a positive association is shown between degree centrality and citation counts (U = 2738017, p = 0.00) where

**Table 6. Summary of association testing results.**

| | T1 | | T2 | | T3 | |
|---|---|---|---|---|---|---|
| | **Cit** | **Doc** | **Cit** | **Doc** | **Cit** | **Doc** |
| Ego-Density (1) | | + | | + | | |
| Degree (2) | | | + | + | + | + |
| Betweenness (3) | | | + | + | + | + |
| Closeness (4) | | + | + | + | + | - |
| Efficiency (5) | + | + | + | + | + | + |
| Constraint (6) | | + | + | | - | |
| Tie Strength (7) | + | + | + | + | | |

+Positively association

-Negative association

authors with numerous collaborative relationships having higher citation counts (Mdn = 2656) than those with fewer relationships (Mdn = 2347). In contrast, authors with numerous collaborative relationships in $t_3$ had lower research productivity (Mdn = 2404) than those with fewer relationships (Mdn = 2594); U = 2887576, p = 0.00. Therefore, we can infer that individual collaborative relationships are no longer effective in the last decade in enhancing research performance compared to the periods of stakeholder theory origin and development ($t_1$ and $t_2$) that required joint efforts to advance the field.

Regarding betweenness centrality and research performance, the results show that authors that lie on the shortest path between other authors had better research performance in $t_2$ in terms of research productivity (Mdn = 1939), U = 1704, p = 0.00; and citation counts (Mdn = 1623), U = 20655, p = 0.00, than those who are not considered intermediaries (Mdn = 969, Mdn = 979 respectively). Similar results are shown in $t_3$ between the low betweenness group in terms of research productivity (Mdn = 2445), U = 119157, p = 0.00; and citation counts (Mdn = 2463), U = 586781, p = 0.00; and the high betweenness group (Mdn = 4585, Mdn = 3897 respectively). The absence of a positive association in $t_1$ can be explained by the low number of authors (n = 156) that disabled the formation of large cliques, in which its structures prompt brokerage salience.

With respect to closeness centrality, the overall results show a positive association in all periods, where authors with low closeness centrality in $t_1$ had lower research productivity (Mdn = 75) that those with high closeness centrality (Mdn = 83), U = 2658, p = 0.04). In $t_2$, the results show that authors with low closeness centrality had low research productivity (Mdn = 978) and citation counts (Mdn = 922) than those with high closeness centrality (Mdn = 1010, 1042 respectively; U = 445944, p = 0.05 for research productivity, U = 405711, p = 0.00 for citation counts. A positive association is observed in $t_3$ regarding citation counts between low closeness group (Mdn = 2286) and high closeness group (Mdn = 2727); U = 2572243, p = 0.00. The only exception is in $t_2$ with research productivity where a negative association is observed where low closeness group having higher research productivity (Mdn = 2525) than the low closeness group (Mdn = 2474); U = 3058094, p = .037. Hence, the findings infer that the closeness of authors to each other, (i.e. being separated by few network steps) was important for all periods in enhancing research performance except for research productivity in $t_3$ which relied more on authors with high degree and betweenness centrality as shown by the above results.

Efficiency is positively associated with research productivity and citation counts for all periods. For $t_1$, authors who were surrounded by non-redundant ties had higher citation counts

(Mdn = 89) and research productivity (Mdn = 80) than those who have a less efficient network position (Mdn = 63 and Mdn = 76, respectively); U = 1977, p = 0.00 for citation counts, U = 2742, p = 0.05 for research productivity. Similarly, authors who were surrounded by non-redundant ties had higher citation counts (Mdn = 1052) and research productivity (Mdn = 1015) than those who have a less efficient network position (Mdn = 929, Mdn = 977 respectively); U = 429965, p = 0.00 for citation counts, U = 472013 p = 0.01 for research productivity. Similarly, efficient authors had higher citation counts (Mdn = 2548) and research productivity (Mdn = 2722) than those who were less efficient (Mdn = 2387, Mdn = 2209 respectively); U = 2848639, p = 0.00 for citation counts, U = 2414066, p = 0.05 for research productivity. These findings indicate that authors surrounded by structural holes–being connected to a primary co-author in a group and receiving novel information–had good research performance. Moreover, it can be argued that expansion of the STM field relied on novel information flowing between efficient authors of different disciplines.

The findings show that constraint is positively associated with research performance in $t_1$ and $t_2$ while in $t_3$ a negative association is shown instead. In particular, authors with redundant ties had higher research productivity in $t_1$ (Mdn = 83; U = 2658, z = -2.8, p = .004) and citation counts in $t_2$ (Mdn = 1028; U = 469269, z = -2.2, p = .03) than those that are less constrained (Mdn = 75, Mdn = 970 respectively). This finding contradicts previous research which showed that constraint is negatively associated with research performance before year 2010 [147]. However, in $t_3$, a negative association is shown were highly contrained individuals (i.e. those with redundant ties) had lower citation counts (Mdn = 2275) than those that are less constrained (Mdn = 2716), U = 25726787, p = 0.00). Therefore, research productivity in $t_2$ and citation counts in $t_3$ have been mainly enhanced via authors with redundant relationships that lead back to same group of co-authors. We argue that with the wide expansion of the collaboration network in $t_3$, that had witnessed the emergence of many scholars, it is difficult for authors to establish relationships with all members of a clique, and therefore, must rely on relationships established with primary actors, reflected by the salience of structural holes.

With respect to tie strength, the findings show a positive association with research performance in $t_1$ and $t_2$. With regards to $t_1$, the results show that authors, who had strong relationships with other authors, had better citations (Mdn = 101) and research productivity (Mdn = 83) than those with weaker ties (Mdn = 56, Mdn = 74 respectively). Similarly, in $t_2$, authors with strong ties had higher citations (Mdn = 1269) and research productivity (Mdn = 1064) than those with weak ties (Mdn = 778, Mdn = 945 respectively). Therefore, the theory of "strong ties" [94] in ehancing productivity is supported by our analysis. Strong relationships between co-authors are essential for increasing citation and publication counts.

## Conclusion and implications

This study descriptively analyzed the evolution of research collaboration networks of authors, institutions and countries, in the STM discipline and identified key actors that are leading collaborative works. In addition, this study examined the longitudinal effect of co-authorship networks on research performance by exploring the associations between collaborative social network variables and each of citation counts and research productivity.

The findings of the authors' collaboration network revealed a premature and fragmented network in $t_1$, where collaboration has happened in the form of sub-networks or cliques of closely knitted actors. In $t_2$, the network increased in size by the emergence of mostly single and dyadic authors which further disintegrated the network. In $t_3$, a larger network and a higher number of cliques emerged, with the most prolific actors having a strong brokerage role (betweenness centrality). The overall results show that stakeholder theory has a wide

scope of interpretations and lacks universal consensus on its concepts and frameworks [34, 35, 148, 149].

The findings of the institutional collaboration networks revealed that the collaboration network in $t_1$ is fragmented into several small size cliques controlled mostly by US institutions. In contrast, a wider international collaboration was witnessed in $t_2$, with the emergence of non US-institutions. The results for $t_3$ showed that the most prolific universities (University of Leeds, University of Washington, University of Toronto) did not belong to the same components, therefore, indicating that the collaboration is led by highly central actors disseminated across the entire network.

The collaboration network of countries originated by being uncondensed, fragmented and highly centralised in $t_1$, with only 16 countries where USA and England being the most prolific actors in STM research. The collaboration network became highly dense and less fragmented in $t_2$ with 74 countries joining the scene. A larger network of 141 countries was observed in $t_3$ with high density and less fragmentation.

Regarding the impact of co-authorship networks on research performance, efficiency was found to be the only network measure positively associated with both citation counts and research productivity in all of the three periods (see Table 6), indicating the importance of structural holes in enhancing research performance. In summary, STM research performance is influenced by authors (1) in highly dense collaborative clusters (ego-density), are (2) close to all other actors in the network, (3) efficient (those that present novel research information); (4) constrained by repetitive relationships and (5) that have strong ties with other authors.

This paper contributes to STM reseach by showing the evolvement of the field and the dynamic changes in its structures. The findings demonstrate that STM is indeed a multi-disciplinary discipline, reflected by fragmented co-authorship network from $t_1$ to $t_3$ and the emergence of a high number of single and dyadic author representing disunity in STM research interest. This heeds the growing calls to explore the structural composition of STM [150]. Fig 6 supports this notion which illustrates keyword co-occurrence networks in STM discipline in $t_1$, $t_2$, $t_3$. The main keywords with the highest co-occurrence in $t_1$ are 'stakeholder analysis', 'stakeholder' and 'stakeholder theory', which all were fundamental and related concepts in STM but each belonging to a different clique. This indicates that STM had not received profound universal consensus at that time and had various comprehensions. However, the application of STM in other disciplines was on the rise, especially with 'stakeholder analysis' coinciding with 'strategic planning', 'climate change' and 'participatory research'. In $t_2$, new major keywords appeared such as 'corporate social responsibility', 'business ethics' and 'corporate governance', all belonging to the same cluster (all having a red color) indicating wide acceptance of stakeholder theory as a theory of management and ethics. Other non-related STM keywords ('climate change', 'health', ', 'resource-based view', 'governance', 'networks', etc.) had also emerged, indicating that STM is a "living Wiki" that is continuously growing through the collaboration of stakeholder scholars from different research fields [32].

This study provides practical contributions to scientists in the STM field and educational managements worldwide. First, the concrete findings from the association testing can help stakeholder scientists improve their research performance by altering the configuration of their collaborative relationships, especially degree, betweenness, and closeness centralities. Institutions can benefit from these results to increase citations rates and research productivity. Second, this study provides empirical evidence regarding the structure of collaboration networks and central actors, that if acted upon, can directly or indirectly lead the allocation of government funding, maximization of research outputs, improvement of research community

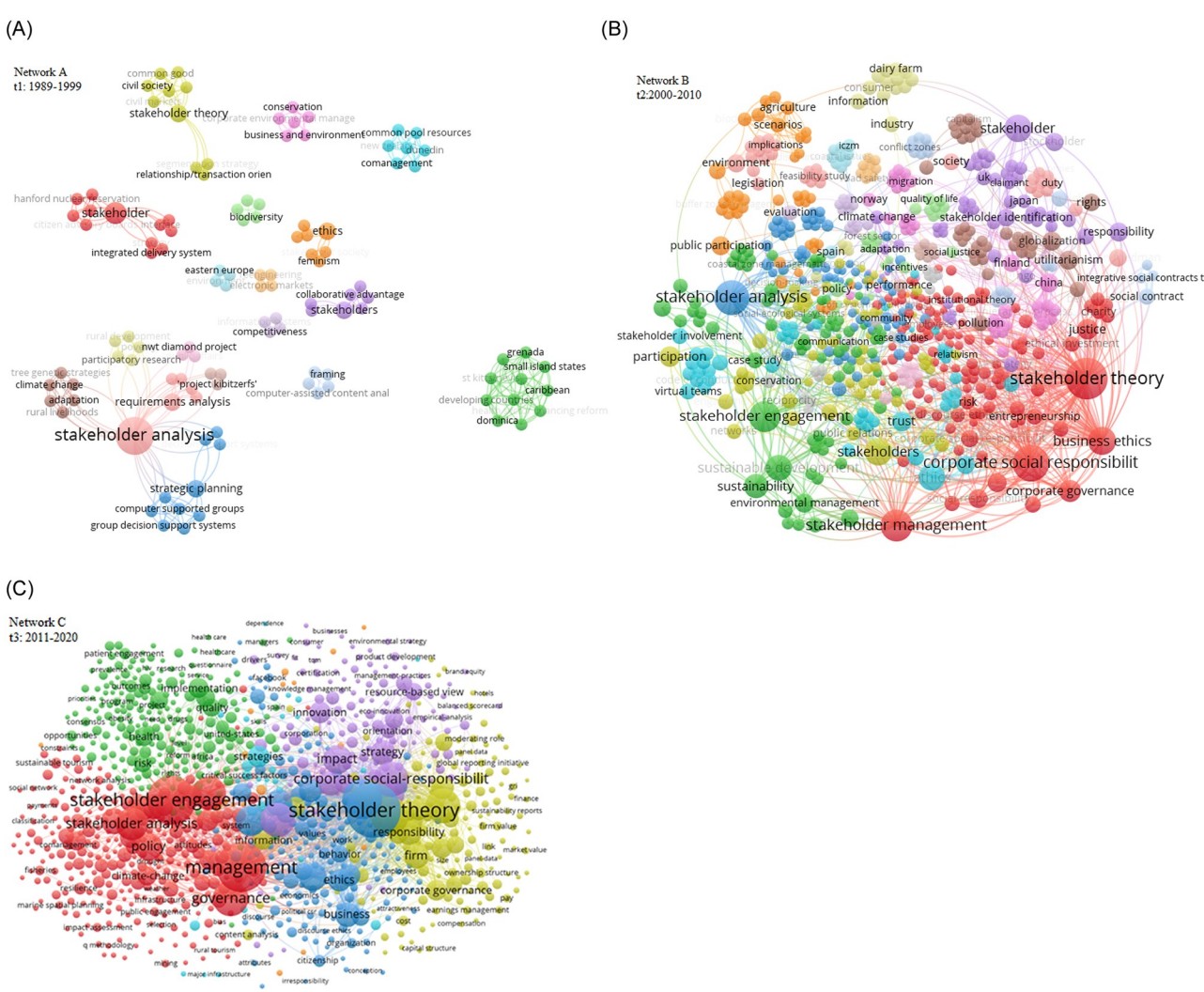

**Fig 6. Co-occurrence network of keywords in STM in $t_1$ (Network A), $t_2$ (Network B) and $t_3$ (Network C).** Each node/circle represents a keyword in STM research. The size of each node size is proportional to the number of connections. A line connecting two nodes indicates an affiliation between two keywords. Node color represents related clusters of keywords.

reputation and the enhancement of cost savings [29, 30], that can all improve collaboration and developing coordinated research programs that can advance the field.

## Supporting information

**S1 File.**
(XLSX)

## Author Contributions

**Conceptualization:** Julian Fares, Kon Shing Kenneth Chung, Alireza Abbasi.

**Methodology:** Julian Fares.

**Supervision:** Kon Shing Kenneth Chung.

**Visualization:** Julian Fares.

**Writing – original draft:** Julian Fares.

**Writing – review & editing:** Julian Fares, Kon Shing Kenneth Chung, Alireza Abbasi.

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
