## [Decision Letter · Decision Letter 0]

30 Dec 2020

PONE-D-20-37035

Understanding Collaboration Networks in Stakeholder Theory and Management: A Longitudinal Approach

PLOS ONE

Dear Dr. Fares,

Thank you for submitting your manuscript to PLOS ONE. After careful consideration, we feel that it has merit but does not fully meet PLOS ONE’s publication criteria as it currently stands. Therefore, we invite you to submit a revised version of the manuscript that addresses the points raised during the review process.

We look forward to receiving your revised manuscript.

Kind regards,

Ghaffar Ali, PhD

Academic Editor

PLOS ONE

Journal Requirements:

2.We note that you have indicated that data from this study are available upon request. PLOS only allows data to be available upon request if there are legal or ethical restrictions on sharing data publicly. For more information on unacceptable data access restrictions, please see http://journals.plos.org/plosone/s/data-availability#loc-unacceptable-data-access-restrictions.

3. Please ensure that you refer to Figure 1 in your text as, if accepted, production will need this reference to link the reader to the figure.

Reviewers' comments:

Reviewer's Responses to Questions

**Comments to the Author**

1. Is the manuscript technically sound, and do the data support the conclusions?

Reviewer #1: Yes

Reviewer #2: Yes

2. Has the statistical analysis been performed appropriately and rigorously? 

Reviewer #1: Yes

Reviewer #2: Yes

3. Have the authors made all data underlying the findings in their manuscript fully available?

Reviewer #1: Yes

Reviewer #2: Yes

4. Is the manuscript presented in an intelligible fashion and written in standard English?

Reviewer #1: Yes

Reviewer #2: No

5. Review Comments to the Author

Reviewer #1: I enjoyed reading the article, the authors presented a longitudinal approach to unveil Collaboration Networks in Stakeholder Theory and Management. I have several comments which must be addressed before considering this article for publications.

English editing of the manuscript is required.

The abstract in the system and manuscript are not similar, please check carefully.

Line 243: UCINET is not explained, full form could be presented on its first appearance in the manuscript.

All abbreviation / acronyms could be thoroughly checked in the revised version.

ED, the ego-density could be mentioned in preceding paragraphs.

Discussion with reference to other studies and findings could have been added to strengthen the inferences derived from the analysis.

Fig 2 the sharp dip in 2020 indicates that authors did not consider the whole year, I would suggest either remove 2020 or add the latest papers as well. Though, there is a possibility of decrease in number of publication due to the pandemic.

Overall, the paper is well written and could be considered for publication after minor revision.

Reviewer #2: 1. This study has some interesting points which carry potential of publication. However, in its current form, some revisions are required in few areas before any final decision. I have few yet significant comments on this paper and analyses. Kindly find my comments and suggestions below:

2. Abstract needs revision in terms of methodology and possible practical policy implications at regional and global levels and in terms of language.

3. Arrange keywords in alphabetical order.

4. Brief history section should be literature review section. And more sophisticatedly arranged.

5. How to validate data or verify because no specific source except WoS provided?

6. Why social network analysis is used? How about other methods?

7. Table captions are always provided at top of the table. Correct all.

8. Conclusions and policy implications is way long section. Reduce it to 1.5 page only please. Only provide the most significant things.

9. Fig. 4, network B, contains very minute words inside. Please use better font size.

10. Try not to use references older than 2012. And also add some citations from this journal PLOS One, to link your paper with this.

11. Lastly, English language editing required. Please avoid use of “we”, “our”, “us” in the paper. Write it in scientific language manner.

6. PLOS authors have the option to publish the peer review history of their article (what does this mean?). If published, this will include your full peer review and any attached files.

Reviewer #1: No

Reviewer #2: No

---

## [Author Response · Author response to Decision Letter 0]

8 Jul 2021

Response to Reviewers

The authors highly appreciate the reviewers’ insightful and helpful comments to improve the manuscript, and they would like to thank them for this great opportunity. Please see below our detailed response to the changes required. All line numbers in this document refer to the manuscript file while track changes are showing.

Editors Comments:

1. Comment: Please ensure that your manuscript meets PLOS ONE's style requirements, including those for file naming. The PLOS ONE style templates can be found at

The manuscript has been revised according to the guidelines listed in the above two links. For instance, see the updated affiliations at the beginning of the manuscript (lines 6-14). Also, the heading font size has been changed to 16 for heading 1, 14 to heading 2, etc. 

2. Comments: We note that you have indicated that data from this study are available upon request. PLOS only allows data to be available upon request if there are legal or ethical restrictions on sharing data publicly. 

The dataset has now been submitted during the second revision and will be available.

3. Comment: In your revised cover letter, please address the following prompts:

The above points have been included in the cover letter where there are no legal restrictions to share the data. 

4. Comment: Please ensure that you refer to Figure 1 in your text as, if accepted, production will need this reference to link the reader to the figure.

Figure 1 is now referred to in the text. Kindly see lines 244-245 (track changes showing). All figures now are referred to in the text.

5. Comment: Please include a separate caption for each figure in your manuscript.

A separate caption is now included for each figure.

Reviewer #1: 

The authors would like to thank Reviewer 1 for the insightful comments. Indeed, the comments helped improved the paper a lot.

1. Comment 1: English editing of the manuscript is required.

The authors agree that English language is wordy in some sections, and therefore, english editing has now been done throughout the manuscript. For instance, kindly see:

The Abstract

Introduction

Lines 365-369

Lines 376-382

Lines 391-394

Lines 405-411

Lines 424-428

Conclusion and Implication part

And other sections of the document.

2. Comment: The abstract in the system and manuscript are not similar, please check carefully.

They are updated now to be the same

3. Comment: Line 243: UCINET is not explained, full form could be presented on its first appearance in the manuscript.

Surprisingly, there is no full form for “UCINET”. UCINET is known as a windows software package for social network analysis (see https://sites.google.com/site/ucinetsoftware/home). The authors now included an explanation for UCINET in the manuscript (see lines 264-266)

4. Comment: All abbreviation / acronyms could be thoroughly checked in the revised version.

The main abbreviations are STM and SNA:

The abbreviation used for “Stakeholder theory and management” is “STM”. Now it has been used to replace the full word in line 33.

The abbreviation used for “social network analysis” is “SNA”. Now it has been used to replace the full word in lines:

114, 122, 179, 263, 264,802

There are no other abbreviations in the text.

5. Comment: ED, the ego-density could be mentioned in preceding paragraphs.

Kindly note that “ego-density” is mentioned in the abstract (line 44), introduction (124), literature (line 179) and in the remaining sections of the paper.

If it is requested that the abbreviation “ED” for “ego-density” to be used instead, the authors would like to kindly note that this can be done, however, they prefer that the full term “ego-density” to be used in conjunction with the other social network measures that have no abbreviations, such as betweenness, closesess, efficiency, etc. Otherwise, the authors have to find an abbreviation to all of these, which perhaps, might make it harder for the reader to memorize all of these terms while reading the paper.

6. Comment: Discussion with reference to other studies and findings could have been added to strengthen the inferences derived from the analysis.

The results of this study are divided into 3 main section – authors, institutions and countries. Therefore, we have now referenced studies that support/contradict our findings for each of the three sections. 

For authors, see: (the numbers of references changes, see again)

Reference number 115, line 394

Reference number 18, line 409

Reference number 116, line 427

For institutions, see:

Reference 16 line 470

Reference 20, line 492

Reference 20, line 503

For countries:

See references 20,142,144 in line 509

7. Comment: Fig 2 the sharp dip in 2020 indicates that authors did not consider the whole year, I would suggest either remove 2020 or add the latest papers as well. Though, there is a possibility of decrease in number of publication due to the pandemic.

The authors would like to thank the reviewers for this unintentionally overlooked mistake. The authors would like to note that they submitted this paper on November 24, and haven’t taken into consideration the published papers in December. After a review of all the papers published in 2020 (see picture below), the number increased from 256 to 356. So the final number is 356. The authors believe that the drop in number from 1163 to 356 is due to the Covid – 19 pandemic, which they personally experienced and has also impacted their publication performance overall as well as that of others.

Reviewer #2:

The authors would like to thank Reviewer 2 for the insightful comments. Indeed, the comments helped improved the paper a lot.

1. Comment: Abstract needs revision in terms of methodology and possible practical policy implications at regional and global levels and in terms of language.

These changes have now been made:

For revision in terms of methodology, see lines 28-35 (track changes showing)

For revision in terms of practical policy, see lines 45-53

Language has been improved by reducing the number of words and using more straightforward terminologies across the entire abstract, and also across the entire manuscript. 

2. Comment 3. Arrange keywords in alphabetical order

Done

3. Comment: Brief history section should be literature review section. And more sophisticatedly arranged.

“Brief history” is now replaced by” Literature Review”.

The subsection “Origins of STM” has been included below “Literature Review”

New subsection “Expansion of STM” has been included with new ideas about stakeholder analysis expansion (lines 151 and 155). 

Subsection “social network Theories” has been included within the “literature review” section

4. Comment: How to validate data or verify because no specific source except WoS provided?

The authors would like to kindly note that data, in such form of research, cannot be validated using traditional research validation techniques. The data is obtained from Web of Science (WoS) which is a trusted public source and is widely regarded by scholars as fairly comprehensive and robust. The same approach used in this study can be used to extract data from other scholarly databases like Scopus, Dimensions etc. However, there is overlaps among such data sources and WoS is the pioneer and most trusted source for meta-data analysis of academic articles although it is coverage may be less than other, focusing mainly on prominent journals (publishers).

5. Comment: Why social network analysis is used? How about other methods?

The reason why social network analysis research has been used in this study is because relationships are of paramount importance in explaining behavior, and in particular, how research performance is affected by the structure of authors’ relationships. We rely on a “networks perspective” that uses individual relations to explain individual outcomes. Also, social network analysis has been the main tool for exploring scientific collaboration networks in many studies (kindly see reference numbers 1,2,4,6,12,18,20 etc)

Regarding other methods, there is a wealth of social network models that can be utilized, but the choice boils down to which method addresses the papers’ aims: 

(i) explore the evolution of research collaboration networks, (ii) identify the key actors (authors, institutions and countries) that are leading collaborative works in each sub-period, and (iii) understand the longitudinal effect of co-authorship networks on research performance

These aims require social network analysis to be used in order to quantify the structure of relationships

There are many social network methods in literature, and therefore, the authors have identified a number of well-known models that have been used but cannot be operationalised for this study. Among the popular network models is Exponential Random Graph Models (ERGM) that explores if a network deviates significantly from chance. ERGM requires that the social network is treated as the dependent variable, where in our study, the network constructs were treated as the independent variables and research performance as the dependent variables. Nevertheless, ERGMs requires incorporate nodal attributes in model estimation, where in our study, the characteristics of scientists were excluded. With respect to models that treat time more explicitly in the sense that they model what happens at the micro (individual) level at each point in time is network exposure which explores the social influence in diffusion, a topic that also falls out of scope. Kindly note that there are also other models that cannot be operationalized in this study, but the authors are afraid that discussing them would diverge the study away from its objective.

5. Comment: Table captions are always provided at top of the table. Correct all.

Done now for all 6 Tables.

6. Comment: Conclusions and policy implications is way long section. Reduce it to 1.5 page only please. Only provide the most significant things.

The authors reduced the number pages from 6 to less than 2.5 pages. The authors would like to kindly ask, if possible, to have 2.5 pages instead of 1.5, because we have too many results that need to be summarized (author, institution, countries, association testing) and there is also the theoretical and practical implications that need to be included as well.. Because it is a long manuscript so to speak, and has different research objectives, its better to have a conclusion that discusses briefly each objective:

the evolution of collaboration networks for authors, instituions and countries.

associated testing between network variables and research performance.

we also include a paragraph about key words which is important, and associated figures.

practical contribution.

7. Comment: Fig. 4, network B, contains very minute words inside. Please use better font size.

Larger font size is used now. Old figure deleted.

8. Comment: Try not to use references older than 2012. And also add some citations from this journal PLOS One, to link your paper with this.

The authors would like to kindly note that since we are covering an era between 1989 and 1999, and between 2000 and 2010, we had to include studies published in that era. However, these have been minimized now, and other new references, related to PLOS one and other journals, have been included/

References added (some are PLOS ONE)

Jappe A. Professional standards in bibliometric research evaluation? A meta-evaluation of European assessment practice 2005–2019. PloS one. 2020;15(4):e0231735.

Walker, J., Chaar, B. B., Vera, N., Pillai, A. S., Lim, J. S., Bero, L., & Moles, R. J. (2017). Medicine shortages in Fiji: A qualitative exploration of stakeholders’ views. PLoS One, 12(6), e0178429.

Naseem MA, Lin J, Rehman RU, Ahmad MI, Ali R. Moderating role of financial ratios in corporate social responsibility disclosure and firm value. PloS one. 2019;14(4):e0215430

Jensen MC. Value maximisation, stakeholder theory and the corporate objective function. Unfolding Stakeholder Thinking: Routledge; 2017. p. 65-84. Sartas, M., Van Asten, P., Schut, M., McCampbell, M., Awori, M., Muchunguzi, P., ... & Leeuwis, C. (2019). Factors influencing participation dynamics in research for development interventions with multi-stakeholder platforms: A metric approach to studying stakeholder participation. PloS one, 14(11), e0223044.

LeClair, A. M., Kotzias, V., Garlick, J., Cole, A. M., Kwon, S. C., Lightfoot, A., & Concannon, T. W. (2020). Facilitating stakeholder engagement in early stage translational research. PloS one, 15(7), e0235400.

Zweigenthal, V. E., Pick, W. M., & London, L. (2019). Stakeholders’ perspectives on Public Health Medicine in South Africa. PloS one, 14(8), e0221447

Political stakeholder theory: The state, legitimacy, and the ethics of microfinance in emerging economies. Business Ethics Quarterly. 2017;27(1):71-98.

Freeman RE, Dmytriyev S. Corporate social responsibility and stakeholder theory: Learning from each other. Symphonya Emerging Issues in Management. 2017(1):7-15.

Old References removed:

Katz JS, Martin BR. What is research collaboration? Research policy. 1997;26(1):1-18.

De Haan J. Authorship patterns in Dutch sociology. Scientometrics. 1997;39(2):197-208.

Smith HJ. The shareholders vs. stakeholders debate. MIT Sloan Management Review. 2003;44(4):85-90.

Wallace JS. Value maximization and stakeholder theory: compatible or not? Journal of Applied Corporate Finance. 2003;15(3):120-7.

Zingales L. In search of new foundations. The journal of Finance. 2000;55(4):1623-53.

Orts EW, Strudler A. The ethical and environmental limits of stakeholder theory. Business Ethics Quarterly. 2002:215-33.

Laband DN, Tollison RD. Intellectual collaboration. Journal of Political economy. 2000;108(3):632-62.

Vučković-Dekić L. Authorship-coauthorship. Archive of Oncology. 2003;11(3):211-2.

Berman SL, Wicks AC, Kotha S, Jones TM. 1999. Does stakeholder orientation matter? The relationship between stakeholder management models and firm financial performance. Academy of Management journal

Jones TM. Instrumental stakeholder theory: A synthesis of ethics and economics. Academy of management review. 1995;20(2):404-37.

Jones TM, Wicks AC. Convergent stakeholder theory. Academy of management review. 1999;24(2):206-21.

Borgatti SP. Centrality and AIDS. Connections. 1995;18(1):112-4

Boiko PE, Morrill RL, Flynn J, Faustman EM, Belle Gv, Omenn GS. Who holds the stakes? A case study of stakeholder identification at two nuclear weapons production sites. Risk Analysis. 1996;16(2):237-49

Chubin D, Friedman R, Kemp K, Fainberg A, Linsenmeyer J. Policy analysis at the US Office of Technology Assessment. International Journal of Technology Management. 1996;11(5-6):589-603.

Harrison JS, Freeman RE. Stakeholders, social responsibility, and performance: Empirical evidence and theoretical perspectives. Academy of management Journal. 1999;42(5):479-85.

9. Comment: Lastly, English language editing required. Please avoid use of “we”, “our”, “us” in n

English language has been edited throughout the paper and all “We”s have been removed (see lines: 25, 30, 111, 119, 126, 223, 282, 312, etc.

To see the English editing in some sections, kindly see below:

Abstract

Introduction

Lines 365-369

Lines 376-382

Lines 391-394

Lines 405-411

Lines 424-428

Conclusion and Implication part

And other sections of the document.

---

## [Decision Letter · Decision Letter 1]

22 Jul 2021

Stakeholder Theory and Management: Understanding Longitudinal Collaboration Networks

PONE-D-20-37035R1

Dear Dr. Fares,

We’re pleased to inform you that your manuscript has been judged scientifically suitable for publication and will be formally accepted for publication once it meets all outstanding technical requirements.

Kind regards,

Ghaffar Ali, PhD

Academic Editor

PLOS ONE

Additional Editor Comments (optional):

Reviewers' comments:

Reviewer's Responses to Questions

**Comments to the Author**

1. If the authors have adequately addressed your comments raised in a previous round of review and you feel that this manuscript is now acceptable for publication, you may indicate that here to bypass the “Comments to the Author” section, enter your conflict of interest statement in the “Confidential to Editor” section, and submit your "Accept" recommendation.

Reviewer #1: All comments have been addressed

Reviewer #2: All comments have been addressed

2. Is the manuscript technically sound, and do the data support the conclusions?

Reviewer #1: Yes

Reviewer #2: Yes

3. Has the statistical analysis been performed appropriately and rigorously? 

Reviewer #1: Yes

Reviewer #2: Yes

4. Have the authors made all data underlying the findings in their manuscript fully available?

Reviewer #1: Yes

Reviewer #2: Yes

5. Is the manuscript presented in an intelligible fashion and written in standard English?

Reviewer #1: Yes

Reviewer #2: Yes

6. Review Comments to the Author

Reviewer #1: I am now happy with the revised version. The authors have revised the manuscript and all the comments have be addressed.

Reviewer #2: Authors have made significant changes and responded to all comments and suggestions well. I am glad to pass on my recommendations.

Cheers!

7. PLOS authors have the option to publish the peer review history of their article (what does this mean?). If published, this will include your full peer review and any attached files.

Reviewer #1: No

Reviewer #2: No

---

## [Editor Report · Acceptance letter]

4 Oct 2021

PONE-D-20-37035R1 

Stakeholder Theory and Management: Understanding Longitudinal Collaboration Networks 

Dear Dr. Fares:

I'm pleased to inform you that your manuscript has been deemed suitable for publication in PLOS ONE. Congratulations! Your manuscript is now with our production department. 

Kind regards, 

on behalf of

Prof. Ghaffar Ali 

Academic Editor

PLOS ONE